# Advancing Peptide-Based Vaccines Against *Candida*: A Comparative Perspective on Liposomal and Synthetic Formulations

**DOI:** 10.3390/jof11100715

**Published:** 2025-10-02

**Authors:** Hong Xin

**Affiliations:** Department of Microbiology, Microbiology, Immunology & Parasitology (MIP), LSU Health Sciences Center, New Orleans, LA 70112, USA; hxin@lsuhsc.edu

**Keywords:** *Candida auris*, peptide vaccine, SNAP platform, CoPoP liposomes, antifungal immunotherapy

## Abstract

The growing threat of multidrug-resistant fungal pathogens, especially *Candida auris*, has underscored the need for effective antifungal vaccines. This commentary highlights recent advances in peptide-based vaccination using the SNAP (Spontaneous Nanoliposome Antigen Presentation) platform, focusing on the FM-SNAP vaccine, a bivalent liposomal formulation targeting the surface-expressed peptides fructose bisphosphate aldolase (Fba) and methionine synthase (Met6). Compared to earlier constructs such as MP12, FM-SNAP achieves superior immunogenicity and long-lasting protection at lower antigen doses. It elicits balanced Th1/Th2 cytokine responses and demonstrates durable efficacy in both immunocompetent and complement-deficient mouse models. The platform’s compatibility with clinically approved adjuvants (MPLA and QS-21), modular peptide design, and potential for multi-pathogen applications underscores its translational promise. FM-SNAP exemplifies a next-generation vaccine strategy that is both scalable and adaptable for high-risk immunocompromised populations.

## 1. Introduction

Despite decades of research and alarming increases in drug-resistant fungal infections, no licensed vaccine exists for human use against *Candida* species, or any fungal pathogen. The 2024 study by Huang et al. [1] represents a major advance in peptide vaccine development against disseminated candidiasis. Building on Xin group’s prior work with the MP12, which is synthetic conjugate peptide Fba-Met6 vaccine, the authors utilize a novel liposomal platform SNAP (Spontaneous Nanoliposome Antigen Particle) to display two conserved *Candida* cell surface peptides Fba and Met6. The results demonstrate improved immunogenicity, enhanced protection, and efficient antigen delivery in murine models of *Candida auris* and *Candida albicans* invasive infections. This commentary highlights the immunological significance of this new formulation, and considers the broader implications of the SNAP platform in developing scalable, pan-Candida synthetic peptide vaccines.

## 2. Background: Candida Immunity and Vaccine Challenges

*Candida* species are common fungal commensals capable of causing opportunistic infections that range from superficial mucosal candidiasis to life-threatening systemic bloodstream infections [2]. Among them, *C. auris* has emerged as a critical global health threat due to its high transmissibility, persistence in healthcare settings, and multidrug resistance to all three major antifungal classes [3]. In recognition of the growing burden of fungal disease, the World Health Organization (WHO) recently classified both *Candida auris* and *Candida albicans* as priority fungal pathogens, with *C. auris* ranked as the first-ever fungal pathogen in the WHO’s critical priority group [4]. This designation underscores the urgent need for antifungal vaccines and immunotherapies as part of a global response to drug-resistant fungal infections.

## 3. The SNAP Platform: A Technological Leap

The SNAP (Spontaneous Nanoliposome Antigen Presentation) platform represents a transformative advancement in peptide vaccine delivery. It enables the non-covalent anchoring of polyhistidine-tagged peptides onto cobalt-porphyrin-phospholipid (CoPoP) liposomes, allowing for spontaneous, stable, and multivalent surface display of antigens [2]. This modular approach preserves the native conformation and epitope accessibility of peptide antigens while eliminating the need for chemical conjugation, which can compromise immunogenicity or scalability. In addition to its innovative antigen-display mechanism, the SNAP system incorporates two well-established adjuvants, monophosphoryl lipid A (MPLA) and QS-21, both of which are included in licensed human vaccines such as *Shingrix* vaccine, which protects against herpes zoster (shingles). MPLA and QS-21 are also used in other licensed vaccines, such as Cervarix (for HPV) and in a malaria vaccine, highlighting their broad utility as adjuvants. This improves the translational feasibility of SNAP-based formulations by building on the known safety and immune-enhancing properties of these adjuvants.

In the context of antifungal vaccines, Huang et al. (2024) [1] demonstrated that both monovalent formulations, F-SNAP (targeting Fba) and M-SNAP (targeting Met6), elicited strong IgG responses and induced mixed Th1/Th2 cytokine profiles, including IL-2, IFN-γ, and TNF-α. However, the bivalent FM-SNAP formulation, which simultaneously displays both Fba and Met6 epitopes, yielded superior immunogenicity and broader protection. This enhanced efficacy likely results from cooperative antigen presentation and T cell activation, along with the immunodominance and surface accessibility of both peptide targets across multiple *Candida* species. Taken together, these features position SNAP as a powerful, scalable platform for peptide-based vaccines targeting not only fungal pathogens but also bacterial and viral agents in high-risk populations (Figure 1).

## 4. Immunological Comparison: FM-SNAP vs. MP12

The MP12 formulation, developed as an earlier-generation vaccine, consists of a fully synthetic, chimeric peptide construct in which the Fba and Met6 epitopes are linked by functional amino acid spacers (e.g., KK, RGD) and a universal T helper epitope (TT947–967), but not conjugated to a protein carrier [5]. This multivalent peptide was delivered intramuscularly with the adjuvant Adjuplex and required a 15 µg total peptide dose to achieve protection against disseminated candidiasis in mice. While MP12 induced protection and generated both specific IgG1 and IgG2a responses, the cytokine profile was relatively modest, with limited induction of polyfunctional T cells. In contrast, the FM-SNAP formulation utilizes cobalt-porphyrin–phospholipid (CoPoP) liposomes to anchor Fba and Met6 peptides via His-tags, allowing for spontaneous and stable multivalent display without chemical conjugation. Using a lower total dose of only 3 µg (1.5 µg of each peptide), FM-SNAP induced robust Th1/Th2-balanced immunity with significantly enhanced IFN-γ, IL-2, and TNF-α production by splenocytes. Importantly, the IgG2a:IgG1 ratio favored Th1 polarization, and FM-SNAP conferred protection in both BALB/c and C5-deficient A/J mice. Collectively, these findings suggest that FM-SNAP improves immunogenicity and protective efficacy while reducing the required antigen dose. Compared to MP12, the liposomal SNAP platform facilitates enhanced antigen presentation, promotes polyfunctional T cell activation, and may offer superior scalability and translational potential.

### Experimental Evidence and Progress

Quantitative comparisons further support the superior performance of FM-SNAP over earlier formulations. In BALB/c mice, FM-SNAP vaccination resulted in 100% survival against lethal *Candida albicans* challenge, compared to 60% with MP12 and 40–60% with monovalent F-SNAP or M-SNAP. FM-SNAP also elicited significantly higher serum IgG2a titers, and cytokine profiling showed increased IL-2, IFN-γ, and TNF-α production from splenocytes. Brain fungal burden was reduced by over 3 log units in FM-SNAP–vaccinated mice compared to controls, emphasizing both systemic protection and organ-specific efficacy. These quantitative data highlight the immunological strength of the bivalent SNAP-based formulation.

Beyond Th1/Th2 cytokine induction and antibody class switching, FM-SNAP may also engage critical T cell subsets important for fungal immunity. In particular, Th17 responses play a vital role in mucosal antifungal defense, and future studies are warranted to determine whether FM-SNAP induces IL-17A-producing CD4+ T cells. Additionally, follicular helper T cells (Tfh) are essential for sustained germinal center reactions and long-lived antibody production, particularly following peptide-based vaccination. Although not directly measured in the current models, the strong humoral memory responses observed suggest that FM-SNAP may also stimulate central memory T cells and Tfh populations, contributing to the observed long-term protection. Future mechanistic work will be necessary to fully characterize these cellular pathways.

## 5. Pan-Candida Potential of the FM-SNAP Vaccine: Immunological Breadth and Translational Promise

The FM-SNAP vaccine formulation targets two highly conserved and surface-expressed peptides, fructose bisphosphate aldolase (Fba) and methionine synthase (Met6), which are consistently present across *Candida albicans*, *Candida auris*, and several other medically relevant *Candida* species. These peptides were rationally selected using bioinformatic tools for their surface accessibility, minimal homology to mammalian proteins, and strong predicted MHC class II binding affinity [6]. Both epitopes have demonstrated immunodominance in murine models and are naturally immunogenic in humans, as evidenced by the detection of Fba- and Met6-specific IgG in patients with invasive candidiasis [6,7]. Notably, these antibody responses were associated with better clinical prognosis, highlighting their potential relevance for protective immunity. Consistently, human IVIG enriched with Fba- and Met6-specific IgGs also provided significant protection in murine models [8].

The FM-SNAP formulation utilizes a cobalt-porphyrin–phospholipid (CoPoP) liposomal delivery platform to anchor His-tagged Fba and Met6 peptides. This enables spontaneous multivalent display of the antigens on the liposome surface without chemical conjugation, preserving native epitope structure while co-delivering potent adjuvants MPLA and QS-21. Compared to earlier vaccine constructs such as MP12, which combines Fba and Met6 using amino acid spacers and a universal T-helper epitope, FM-SNAP achieves superior immune activation using one-tenth the peptide dose (3 µg vs. 15 µg). Importantly, Huang et al. demonstrated that FM-SNAP induces robust IgG2a and IgG1 antibody responses and strong polyfunctional T cell activation characterized by increased IL-2, IFN-γ, and TNF-α production. The vaccine provided significant protection in both immunocompetent BALB/c mice and complement-deficient A/J mice, which represent a clinically relevant high-risk immunocompromised populations for susceptibility to invasive fungal infections. FM-SNAP-elicited immune sera exhibited strong binding to live *Candida auris* yeast cells, reflecting not only the surface accessibility of the vaccine-targeted epitopes but also a robust humoral response characterized by high IgG1 titers. Additionally, the vaccine conferred long-lasting protection, with efficacy sustained even when mice were challenged five months post-vaccination, an indicator of durable immune memory.

Moreover, FM-SNAP demonstrated efficacy against drug-resistant *C. auris* strains and achieved significantly reduced fungal burdens and improved survival in both systemic *C. albicans* and *C. auris* challenge models. These findings underscore its potential as a pan-*Candida* vaccine. Given its safety, scalability, and capacity to elicit both humoral and cellular immunity, FM-SNAP holds promise not only for systemic protection but also for future adaptation to topical or mucosal delivery platforms, particularly in preventing device-associated infections and colonization at barrier sites.

## 6. Next Steps and Broader Applications of the FM-SNAP Vaccine Platform

The promising protective efficacy and robust immune activation observed with FM-SNAP suggest multiple directions for future development, including IL-17RA-deficient and neutropenic mice. While these models represent distinct aspects of immunocompromise—such as impaired Th17 signaling or myelosuppression—they offer important insights into how FM-SNAP may perform in patients with varying degrees of immune dysfunction, including those undergoing chemotherapy or suffering from persistent neutropenia. Demonstrating efficacy in these hosts would broaden the clinical scope of FM-SNAP beyond complement-deficient systems.

While FM-SNAP is primarily designed for systemic immunization, its robust induction of circulating antibodies and Th1/Th2 immune responses may provide protection at mucosal surfaces or barrier sites where fungal colonization occurs. The strong surface-binding capacity of FM-SNAP-elicited antibodies supports their potential use in therapeutic strategies, such as passive immunotherapy or antibody-based prevention of biofilm formation on devices like central venous catheters. To complement systemic vaccination, future studies could explore whether FM-SNAP-induced antibodies translocate to mucosal compartments or whether the platform could be adapted to generate mucosal immunity through alternative delivery routes. However, direct topical or biofilm-disrupting applications are better suited for antifungal monoclonal antibodies or other localized therapeutic agents.

Another important avenue involves combining FM-SNAP with conventional antifungal drugs or therapeutic monoclonal antibodies. Such combination strategies could enhance early fungal clearance, reduce drug resistance emergence, and enable lower antifungal doses in patients with limited tolerance or organ dysfunction. The dual activation of humoral and cellular immunity by FM-SNAP provides a mechanistic basis for synergy with other immunotherapies.

## 7. Multiplexed Vaccine Potential of the SNAP Platform

The SNAP platform’s capacity to stably anchor diverse His-tagged peptides or proteins makes it inherently pathogen-agnostic and highly adaptable. This opens the door to multiplexed vaccine formulations that can incorporate antigens from fungi, bacteria, and viruses. Such versatility is particularly valuable in immunocompromised populations, such as patients undergoing chemotherapy or hematopoietic stem cell transplantation, who are at high risk for polymicrobial infections. These patients often exhibit disrupted mucosal barriers and altered innate immunity, creating a permissive microenvironment for co-infections with *Candida*, *Aspergillus*, and bacterial pathogens such as *Pseudomonas aeruginosa* and *Staphylococcus aureus* [2,9]. The ability of SNAP to co-display multiple antigens on the same liposome enables the development of broad-spectrum prophylactic vaccines tailored to these high-risk groups. This strategy could reduce reliance on prolonged antifungal or antibacterial drug regimens and lower the risk of resistance emergence. Moreover, the same CoPoP-based liposomal technology used in FM-SNAP has advanced to human clinical trials for SARS-CoV-2 vaccines and is currently in phase 2 evaluation, reinforcing its translational feasibility [10]. This clinical foundation provides a compelling rationale for applying the SNAP platform to multi-pathogen vaccine strategies aimed at protecting the most vulnerable patient populations.

### 7.1. Limitations and Challenges

Although the FM-SNAP platform shows strong preclinical efficacy in BALB/c and A/J mice, murine models have inherent limitations. Unlike humans, mice are not natural hosts for *Candida albicans*, and immune responses—particularly those involving mucosal memory and long-lived plasma cells—may differ across species. Future studies are planned to evaluate FM-SNAP in additional immunocompromised models, including neutropenic and IL-17RA-deficient mice. Long-term goals include GMP manufacturing and advancement to Phase 1 human studies to assess safety, immunogenicity, and dose optimization.

While the FM-SNAP platform demonstrates strong immunogenicity, peptide efficiency, and translational feasibility, several potential limitations should be noted. First, liposomal formulations such as CoPoP-based SNAP vaccines may present stability and storage challenges, particularly under variable environmental conditions or in low-resource settings. Optimization of lyophilization protocols and shelf-life studies will be necessary for broader clinical deployment. Second, widespread clinical use of these vaccines requires improvements in freeze-drying and shelf-life stability. Also, large-scale GMP production may be costly, though the vaccine’s modular design could improve manufacturing consistency. Third, human immune responses are heterogeneous; thus, validation of FM-SNAP across genetically diverse populations, including individuals with varying degrees of immunocompromise, will be important to fully characterize its protective efficacy.

In mouse studies, FM-SNAP has been well tolerated, with no weight loss, behavioral changes, or injection site swelling observed following vaccination. Both adjuvants have established clinical use, supporting the potential safety of the FM-SNAP formulation. While formal GLP toxicology studies are still pending, these initial tolerability data support further development. In addition, the defined, synthetic components of FM-SNAP reduce the risk of off-target immune responses commonly associated with whole-cell fungal vaccines.

### 7.2. Translational Potential and Future Directions

The FM-SNAP formulation uses defined, synthetic peptides and established adjuvants, which streamlines batch reproducibility and regulatory compliance. The CoPoP liposomal technology has already been evaluated in human clinical trials for a SARS-CoV-2 vaccine (EuCorVac-19), demonstrating GMP feasibility. Although results from the Phase 2 trial have not been publicly reported, the platform’s regulatory progression indicates potential for scale-up. Published studies demonstrate that the same formulation induced robust and durable antibody responses in humans, supporting continued development [11]. Future development of FM-SNAP will require integration of stability optimization, lyophilization protocols, and multi-dose vial testing to facilitate clinical deployment.

## 8. Conclusions

The FM-SNAP peptide vaccine represents a major advance in fungal immunization, combining immunological breadth, dose efficiency, and translational readiness. Its protection in multiple mouse models, strong induction of cellular immunity, and compatibility with the clinically advanced CoPoP platform suggest a clear path toward clinical development. Continued evaluation in mucosal and immunocompromised models, as well as human formulation optimization, will be key to reach its full potential.

## Figures and Tables

**Figure 1 jof-11-00715-f001:**
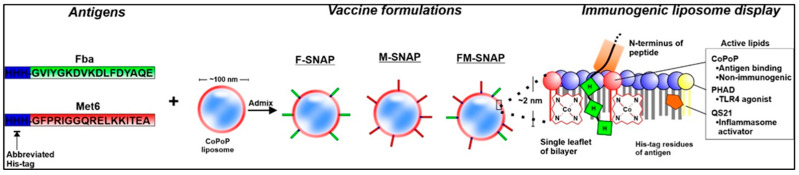
**Schematic illustration of the SNAP vaccine platform**. His-tagged peptide antigens derived from *Candida* surface proteins—Fba (green) and Met6 (red)—spontaneously bind to cobalt-porphyrin-phospholipid (CoPoP) liposomes via metal chelation, forming a stable, biostable nanoparticle vaccine. CoPoP serves as the anchoring lipid and forms the structural basis of the SNAP (Spontaneous Nanoliposome Antigen Presentation) platform. The liposome bilayer is co-formulated with MPLA (a TLR4 agonist) and QS-21 (an inflammasome activator), both of which are adjuvants used in licensed human vaccines such as Shingrix. The resulting nanoparticle displays the peptides multivalently on the outer leaflet of the liposome, preserving epitope conformation and enhancing antigen presentation. FM-SNAP includes both Fba and Met6 epitopes, while F-SNAP and M-SNAP are monovalent formulations. This design supports antigen-specific B and T cell activation and enhanced protection in murine models of disseminated candidiasis.

## Data Availability

Not applicable. No new datasets were generated or analyzed for this commentary.

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
