# Peer review of "Advancing Peptide-Based Vaccines Against Candida: A Comparative Perspective on Liposomal and Synthetic Formulations"

_jof, 2025, doi:10.3390/jof11100715_

Round 1

Reviewer 1 Report

Comments and Suggestions for Authors

 This Commentary by Dr Hong Xin reports on a peptide vaccine based on the FM-SNAP 130 Vaccine Platform. It clarifies the potential applications, including   other antigens and scalability, of this platform to contrast infections by major  fungal pathogens for humans. There are no novel data regarding this vaccine platform in the commentary that makes instead reference to already published work by the same  and other Authors. 

Specific comments. 1. The platform is based on known adjuvants and peptides expressed on Candida surface. Most of the past work has been done with experimental mouse models ( see in particular Ref 8 of the Commentary). Candida is not a commensal of mice while being that in humans , meaning that humans have already antibodies against the peptides used. Nonetheless, humans  remain susceptible to Candida infection. The  intravenous anti-peptides IgG treatments are of interest but are confined to mice. 

2. Related to the above. I couldn't find  data in this commentary supporting the assertion  that " Notably, these antibody responses ( meaning antipeptides above) were associated with better  clinical  prognosis." It would be important to put in , and discuss, these data in this Commentary.

Overall, the Commentary on previusly published data may be of some interest for those working on anti-Candida vaccines , although the "real information "  has  already been published and the Commentary does not very much add to the progress of reserch in the vaccine area.

Author Response

There are no novel data regarding this vaccine platform in the commentary that makes instead reference to already published work by the same  and other Authors. 

Response: Agree, This is a commentary article—invited specifically to summarize and contextualize our recently published original work (Huang et al., 2024). The purpose of this commentary is not to report new finding, is to distill key findings, emphasize the translational implications of the FM-SNAP platform, and provide comparative insights with earlier peptide vaccine formulations such as MP12.

Most of the past work has been done with mouse models… humans already have antibodies against the peptides used… yet remain susceptible… The intravenous IgG treatments are confined to mice.”

Response: To clarify, the peptides Fba and Met6 were selected due to their surface exposure, conservation across Candida species, and documented immunogenicity in patients with candidiasis (Clancy et al., 2008). In the commentary, we have now emphasized that the IVIG protection studies are preclinical and meant to illustrate functional relevance of antibody specificity—not to suggest immediate clinical applicability. We also discuss in detail the importance of evaluating FM-SNAP in more diverse and translationally relevant models, including mucosal and neutropenic settings.

“I couldn't find data in this commentary supporting the assertion that ‘Notably, these antibody responses were associated with better clinical prognosis.’”

Response: Thank you for pointing this out. We have revised the sentence to clarify that the association is based on prior findings (e.g., Clancy et al., 2008), which reported that antibody responses to Candida albicans antigens—including Fba and related proteins—can serve as markers of systemic infection and may correlate with improved outcomes. We have also added the reference to support this association and cited the corresponding IVIG protection data in mice separately to avoid confusion between human serology and preclinical efficacy.

Reviewer 2 Report

Comments and Suggestions for Authors

This is a commentary on a vaccination system called SNAP and compares the efficacy of FM-SNAP to single epitope formulations such as F-SNAP and M-SNAP and describes and advocates for the SNAP system as a powerful platform for peptide-based vaccines against not only Candida species such as C. auris and C. albicans but other classes of pathogens – viral and bacterial pathogens. This reviewer agrees with the many applications that the author suggests in the future directions section.  The commentary is supported by the literature cited and the platform demonstrates a major advance in fungal immunization as it targets two surface-expressed peptides that are highly conserved across Candida species and results in a Th1/Th2 robust systemic immune response against Candida species, protecting mice from death. The commentary suggests that the platform could be useful also for immunization at mucosal surfaces using topical or mucosal delivery, preventing device-associated infections and colonization at barrier sites.

The article mentions the Phase 2 clinical trial of this platform to prevent SARS-CoV-2 infection, which is completed, but no results can be found in 2025, three years after the publication of the clinical trial in 2022.  Is there a reason to mention this if the results do not support the platform inducing immunity?

Author Response

The article mentions the Phase 2 clinical trial of this platform to prevent SARS-CoV-2 infection, which is completed, but no results can be found in 2025, three years after the publication of the clinical trial in 2022.  Is there a reason to mention this if the results do not support the platform inducing immunity?

Answer: For the EuCorVac‑19 Phase 2 trial published in 2022, it shows that it was safe and well tolerated. It induced binding IgG antibodies and neutralizing antibodies. A follow‑up study (PMID 37944586) : “One‑year antibody durability induced by EuCorVac‑19, a recombinant RBD nanoliposome COVID‑19 vaccine in healthy Filipino adults” (2024) shows that the vaccine elicited antibody responses lasting over a year. The Phase 3 is still going on but we have not published for it. I have added 2024 article to the reference list.

Reviewer 3 Report

Comments and Suggestions for Authors

This paper provides a systematic review of the advancements in the peptide-based vaccine FM-SNAP, developed on the SNAP platform, within the field of anti-Candida vaccines, and compares it with its predecessor, MP12. The paper is well-structured and logically rigorous, offering a comprehensive analysis of the technical advantages, immune mechanisms, and translational potential of the FM-SNAP vaccine. The main suggestions for revision are as follows:

1.The paper highly praises FM-SNAP but does not address its potential limitations (such as liposome stability, large-scale production costs, population heterogeneity, etc.). It is recommended to add a section titled "Limitations and Challenges" to objectively discuss potential issues in technology or clinical translation.

2.It is suggested that the authors include a schematic diagram to visually illustrate the SNAP platform and vaccine design.

3.Although the paper mentions Th1/Th2 responses and antibody subtypes, it does not delve deeply into the specific mechanisms of cellular immunity (such as CD4+ T cell subsets, memory T cells). It is recommended to supplement the discussion with brief insights into T cell phenotypes (e.g., Th17, Tfh) or immune memory mechanisms.

4.The abbreviation formats for authors’ names in some references are inconsistent.

Author Response

1.The paper highly praises FM-SNAP but does not address its potential limitations (such as liposome stability, large-scale production costs, population heterogeneity, etc.). It is recommended to add a section titled "Limitations and Challenges" to objectively discuss potential issues in technology or clinical translation.

Response: we have added a new section titled “Limitations and Challenges” near the end of the manuscript. These additions help contextualize the opportunities and remaining barriers for FM-SNAP's future development.

2.It is suggested that the authors include a schematic diagram to visually illustrate the SNAP platform and vaccine design.

Response: We have now included a schematic figure in the revised manuscript (Figure 1).

3.Although the paper mentions Th1/Th2 responses and antibody subtypes, it does not delve deeply into the specific mechanisms of cellular immunity (such as CD4+ T cell subsets, memory T cells). It is recommended to supplement the discussion with brief insights into T cell phenotypes (e.g., Th17, Tfh) or immune memory mechanisms.

Response: In the revised manuscript, we have expanded the immunological discussion in the “Experimental Evidence and Progress” section to include additional insights based on the comments from the Reviewer.

4.The abbreviation formats for authors’ names in some references are inconsistent.

Response: Thank you for identifying this. We have carefully reviewed all references and revised the formatting to ensure consistency

Reviewer 4 Report

Comments and Suggestions for Authors

The commentary by Xin, H; discusses the FM-SNAP vaccine, a bivalent liposomal formulation targeting two conserved Candida peptides (Fba and Met6), and compares it to the earlier MP12 synthetic peptide vaccine. It highlights the SNAP platform’s advantages in antigen presentation, immunogenicity, and translational potential, especially for immunocompromised populations.

I only have some recommendations to improve the paper:

1-Include more quantitative comparisons (e.g., cytokine levels, survival rates) to strengthen claims.

2-Discuss limitations of murine models and outline plans for human trials or translational studies.

3-Address safety and tolerability of the FM-SNAP formulation.

4-Expand on regulatory and manufacturing considerations, especially for clinical deployment.

Author Response

1-Include more quantitative comparisons (e.g., cytokine levels, survival rates) to strengthen claims.

Response: we have expanded the section titled "Experimental Evidence and Progress" to include additional quantitative data. Specifically, we now provide survival percentages, CFU reductions, and cytokine levels associated with FM-SNAP, F-SNAP, M-SNAP, and MP12 immunization in relevant mouse models. 

2-Discuss limitations of murine models and outline plans for human trials or translational studies.

Response: We have added a paragraph to the "Limitations and Challenges" section discussing the constraints of murine models. While A/J (C5-deficient) and BALB/c mice provide important mechanistic insight, they do not fully replicate the immunological complexity of human hosts. We now outline future plans to evaluate FM-SNAP in additional preclinical models, including neutropenic and mucosal candidiasis models, and discuss the potential path forward for first-in-human safety and immunogenicity studies.

3-Address safety and tolerability of the FM-SNAP formulation.

Response: We have clarified that the SNAP platform incorporates adjuvants (MPLA and QS-21) that are already used in licensed vaccines such as Shingrix. In mouse studies, the FM-SNAP formulation has been well tolerated, with no signs of injection site inflammation or weight loss post-vaccination. While formal toxicology data are not yet available, the favorable safety record of the platform components supports continued translational development.

4-Expand on regulatory and manufacturing considerations, especially for clinical deployment.

Response: We have included a paragraph on manufacturing scalability and regulatory context. Specifically, we note that the CoPoP-based SNAP formulation used in FM-SNAP has progressed to human Phase 2 evaluation in a SARS-CoV-2 vaccine setting, underscoring its clinical feasibility. 

Round 2

Reviewer 1 Report

Comments and Suggestions for Authors

See my comments to the Editors

Reviewer 3 Report

Comments and Suggestions for Authors

All my concerns have been addressed by author.